# Insecticidal Activity of Lectin Preparations from *Moringa oleifera* Lam. (Moringaceae) Seeds Against *Alphitobius diaperinus* (Panzer) (Coleoptera: Tenebrionidae)

**DOI:** 10.3390/plants14040511

**Published:** 2025-02-07

**Authors:** Nataly Diniz de Lima Santos, Patryck Érmerson Monteiro dos Santos, Thiago Lucas da Silva Lira, Aline Rodrigues da Silva Santos, Juliane Nancy de Oliveira Silva, Alícia Natalie Silva dos Santos, Marcus Mello Rego de Amorim, Mércia Rodrigues Barros, Luana Cassandra Breitenbach Barroso Coelho, Patrícia Maria Guedes Paiva, Thâmarah de Albuquerque Lima, Gustavo Ferreira Martins, Thiago Henrique Napoleão, Emmanuel Viana Pontual

**Affiliations:** 1Departamento de Morfologia e Fisiologia Animal, Universidade Federal Rural de Pernambuco, Recife 52171-030, PE, Brazil; nataly.santos@ufpe.br (N.D.d.L.S.); marcus.mramorim@hotmail.com (M.M.R.d.A.); 2Departamento de Bioquímica, Centro de Biociências, Universidade Federal de Pernambuco, Recife 50670-901, PE, Brazil; patryck.santos@ufpe.br (P.É.M.d.S.); thiago.silvalira@ufpe.br (T.L.d.S.L.); aline.rodriguess@ufpe.br (A.R.d.S.S.); juliane.nancy@ufpe.br (J.N.d.O.S.); alicia.natalie@ufpe.br (A.N.S.d.S.); luana.coelho@ufpe.br (L.C.B.B.C.); patricia.paiva@ufpe.br (P.M.G.P.); thamarah.albuquerque@ufpe.br (T.d.A.L.); 3Departamento de Medicina Veterinária, Universidade Federal Rural de Pernambuco, Recife 52171-030, PE, Brazil; mercia.barros@ufrpe.br; 4Departamento de Biologia, Universidade Federal de Viçosa, Viçosa 50670-901, MG, Brazil; gmartins@ufv.br

**Keywords:** chitin-binding lectin, natural insecticide, midgut damage, poultry farming, lesser mealworm

## Abstract

The lesser mealworm, *Alphitobius diaperinus*, is a widespread pest in poultry farming, causing economic losses and facilitating the spread of pathogens. Current control methods rely heavily on synthetic insecticides, which raise environmental concerns and contribute to resistance. This study investigated the insecticidal potential of *Moringa oleifera* seed preparations, including an aqueous extract (AE), a protein-enriched fraction (PF), and an isolated lectin (WSMoL), against *A. diaperinus*. Contact assays demonstrated that AE and PF reduced adult survival by up to 90% and 100%, respectively, while WSMoL was effective only against larvae, causing 48% mortality. Ingestion assays showed that AE and PF reduced adult survival by 60% and 30%, respectively; impaired diet utilization; and caused significant weight loss. WSMoL exhibited antinutritional effects, including a 94% reduction in trypsin-like activity, but did not cause lethality, although it may impair long-term survival. Midgut histological analysis revealed epithelial disorganization, vacuolization, and nuclear condensation in the treated insects. These findings suggest that *M. oleifera* seed preparations are promising candidates for sustainable pest control, offering both lethal and sublethal effects on *A. diaperinus*. Future research should focus on the development of formulations and long-term impact assessments on pest populations and non-target organisms, paving the way for eco-friendly alternatives in integrated pest management.

## 1. Introduction

The beetle insect *Alphitobius diaperinus* (Panzer, 1797) (Tenebrionidae), known as the lesser mealworm or litter beetle in English and *cascudinho* in Portuguese, is one of the main pests in poultry farming, both in broiler and laying hen operations. It occurs in various poultry facilities in Brazil and worldwide. This insect is responsible for significant economic damage related to poultry production in agroecosystems, as well as sanitary issues. The lesser mealworm can act as a mechanical carrier for fungi, bacteria, and viruses that cause diseases in birds, such as salmonellosis, Gumboro disease, and avian leukosis, and often results in immunosuppressive diseases, hemorrhagic enteritis, malabsorption syndrome, and Marek’s disease [1].

The duration of the life cycle of *A. diaperinus* (egg, 8–11 larval instars, pupa, and adult) is strongly dependent on temperature, food availability, and humidity, making the species highly adapted to the environment of poultry houses. At 30 °C, the median development times (days) for eggs, larvae, and pupae are 4.4, 26.2, and 5.5 days, respectively [2]. The transition from one larval instar to another can occur every 2 days at 30 °C [3]. Under this temperature condition, the period between oviposition and adult hatching is, on average, 37.9 days, and the adults are generally ready to mate between 5 and 8 days after hatching; the adults have a lifespan of about 3 to 6 months under favorable conditions [2,4,5]. A total of 15 days after the first mating, females can lay 200 to 400 eggs every 1–5 days [6].

The confined rearing of poultry has provided an ideal habitat for the growth and population increase of this insect, which reproduces rapidly, primarily due to the lighting conditions and bird density. High population densities of *A. diaperinus* are found in the bedding of broiler chickens and breeders, and even in the feces of commercial laying hens [4,7]. Besides its ability to act as a reservoir for a variety of avian diseases, *A. diaperinus* directly influences feed conversion, affecting the growth and weight gain of the birds which consume the larvae and adults of the insect, instead of balanced feed. Additionally, the rigid exoskeleton of the adult can cause gastrointestinal injuries in birds, leading to secondary infections. The lesser mealworm is also a structural pest of the facilities, as larvae and adults burrow tunnels and nests in cracks in the walls, floor, beams, and electrical systems. The larvae burrow in search of pupation sites, with the tunnels being expanded when the adults emerge [4,6,8].

The control of *A. diaperinus* populations in poultry facilities should involve the integration of various methods, including physical and chemical control strategies for management and monitoring [9]. The application of pyrethroid-based insecticides is typically performed after cleaning and sanitizing the poultry house to reduce insect population densities. However, the intensive use and high doses of synthetic chemical insecticides have led to a high residual presence of these compounds, the establishment of resistance by the insects, and environmental damage [10]. In this context, there is an increased search for new substances derived from natural sources, which can be more environmentally sustainable and can serve as alternatives for rotation or for combined use with other insecticides, thus minimizing the development of resistance [11]. Plant extracts, lectins, and protease inhibitors are potential insecticidal agents for controlling the populations of various insects [12].

Lectins are carbohydrate-binding proteins which are able to recognize monosaccharides, disaccharides, and polysaccharides, as well as oligosaccharide moieties of glycoconjugates [12]. Certain plant lectins have been reported to exhibit insecticidal activity against pests from various insect orders. The insecticidal effects of lectins are attributed to a variable range of mechanisms, including binding to the midgut epithelial cell layer, leading to functional disruption and cell death; the disruption of the peritrophic matrix; the modulation of digestive enzyme activities; the impairment of nutrient absorption; endocytosis; and the elimination of midgut symbiotic microbiota [12,13]. Especially, chitin-binding lectins are considered to be promising insecticidal agents because of their ability to bind to the peritrophic matrix and other insect structures composed of chitin or containing *N*-acetylglucosamine residues [12].

Seeds of *Moringa oleifera* Lam. (Moringaceae) contain a chitin-binding and water-soluble lectin known as WSMoL, which has been studied for its insecticidal properties. Coelho et al. [14] and Agra-Neto et al. [15] reported the larvicidal activity of WSMoL against *Aedes aegypti* (L.) (Diptera: Culicidae), demonstrating that this lectin could stimulate the in vitro activity of proteases, trypsin-like enzymes, and α-amylase from larval gut extracts. Furthermore, the ovicidal effects of WSMoL on *Ae. aegypti* were documented [16]. Eggs treated with this lectin displayed a disrupted exochorionic structure, with the absence of exochorionic cells and their tubercles in multiple regions. After 72 h, the eggs showed considerable surface deformation and degeneration, and it was noted that the lectin had infiltrated into the embryonic body [17]. WSMoL was also able to stimulate the oviposition of *Ae. aegypti*, even at field conditions, by binding to protein receptors in the female legs [18,19]. *Ephestia kuehniella* Zeller (Lepidoptera: Pyralidae) larvae fed on a WSMoL diet showed impaired weight gain and had their protein digestion capacity compromised by more than 90% [20].

Considering the already-documented insecticidal activity of WSMoL, the present study was conducted to answer the following questions: Would contact with crude or purified formulations containing WSMoL be capable of reducing the survival of the larvae and adults of *A. diaperinus*? Can the survival and the feeding preferences of the adult insects be altered when formulations containing WSMoL are mixed with the feeding substrate? Can the ingestion of these formulations modulate the activities of trypsin-like enzymes or cause structural damage in the midgut of adult insects? The experimental design described here answers these questions using WSMoL, either in its isolated form or in formulations containing it, such as the *M. oleifera* seed aqueous extract (AE) and the protein-enriched fraction (PF) derived from it.

## 2. Results

The protein concentration in AE was determined to be 10.5 mg/mL, with a hemagglutinating activity (HA) of 16, resulting in a specific HA of 1.52. In contrast, PF exhibited a protein concentration of 41.0 mg/mL, an HA of 128, and a specific HA of 3.12, indicating a 2.05-fold purification relative to AE. The lectin WSMoL was isolated from PF with a specific HA of 1280, representing an 842-fold purification relative to the extract and a 410-fold purification compared to PF.

The insecticidal assessment against *A. diaperinus* started evaluating the contact (after a short time) effects of AE, PF, and WSMoL on adults and larvae (fourth instar). In these assays, the insects were exposed to a solution of the samples, and the mortality was assessed after 48 h. The data showed that the treatment with AE (Figure 1a; F_5,54_ = 3.128, *p* = 0.0150) or PF (Figure 1b; F_5,54_ = 4.022, *p* = 0.0036), at 2.0–10.0 mg/mL of proteins, killed *A. diaperinus* adults, while WSMoL was not effective in inducing the death of the insects (Figure 1c; F_5,54_ = 1.000, *p* = 0.4267). Concerning the larvae (Figure 1d), exposure to AE (F_2,27_ = 4.574, *p* = 0.0195), PF (F_2,27_ = 589.8, *p* < 0.0001), and also WSMoL (F_2,27_ = 42.12, *p* < 0.0001) resulted in significant toxicity compared to the control, but with different levels of mortality.

AE (Figure 2a; F_3,36_ = 17.32, *p* < 0.0001) and PF (Figure 2b; F_3,36_ = 5.512, *p* = 0.0032) at 2.0, 6.0, and 10.0 mg of proteins/g of diet were insecticidal when ingested by *A. diaperinus* adults, while WSMoL (0.2–1.0 mg/g) failed in inducing insect mortality through this route (Figure 2c; F_5,54_ = 1.304, *p* = 0.2765). When assessing the nutritional parameters, there was a significant reduction in the relative consumption rate of the diet in all treatments with AE (Figure 3a; F_3,36_ = 12.39, *p* < 0.0001), PF (Figure 3b; F_3,36_ = 15.22, *p* < 0.0001), and WSMoL (Figure 3c; F_5,54_ = 5.314, *p* = 0.0007). AE showed a deterrent effect (Figure 3d) ranging from weak to strong (feeding deterrence index, FDI, of 41.37% to 70.79%) while PF (Figure 3e) caused a moderate or strong deterrent effect (FDI of 56.69% to 72.75%). WSMoL (Figure 3f) showed weak or moderate deterrent effects (FDI of 35.71% to 54.32%).

Regarding the effect of preparations containing WSMoL on diet utilization by adult *A. diaperinus*, the relative biomass gain rate was significantly different from the control in all treatments with AE (Figure 4a; F_3,36_ = 30.58, *p* < 0.0001), PF (Figure 4b; F_3,36_ = 52.81, *p* < 0.0001), and WSMoL (Figure 4c; F_5,54_ = 37.25, *p* < 0.0001). Notably, the values were negative, indicating a loss of biomass. Consistent with this, the efficiency in conversion of ingested food was negative in all AE treatments (Figure 4d; F_3,36_ = 13.37, *p* < 0.0001), in PF at 10 mg/g (Figure 4e; F_3,36_ = 4.695, *p* = 0.0079), and in all treatments with WSMoL (Figure 4f; F_5,54_ = 11.92, *p* < 0.0001).

The trypsin-like activity in the midgut extracts from adult insects reared on diets containing AE, PF, and WSMoL for 21 days was also assessed. The insects that ingested AE at 2.0 mg/g did not show enzymatic activity that was significantly different from the control, whereas those exposed to AE at 6.0 and 10.0 mg/g showed reduced trypsin-like activity in a concentration-dependent way (Figure 5a; F_3,12_ = 17.40, *p* = 0.0001) compared to insects from the control. *A. diaperinus* adults that ingested PF (Figure 5b; F_3,12_ = 29.74, *p* < 0.0001) and WSMoL (Figure 5c; F_3,12_ = 34.68, *p* < 0.0001) at all concentrations showed a decrease in trypsin-like activity of more than 94%, or exhibited negligible enzymatic activity, compared to the control.

Finally, the midguts of *A. diaperinus* adults that survived after ingesting AE (10 mg/g), PF (10 mg/g), or WSMoL (1 mg/g) were analyzed for histological alterations. In the midgut of adults from the control, a simple epithelium was observed, composed of columnar digestive cells with an evident brush border (Figure 6a). Compared to the control group, the midgut of insects treated with AE (Figure 6b) showed elongated epithelial cells, suggesting cellular deformation. Insects treated with PF (Figure 6c) exhibited vacuoles in the cytoplasm, intense nuclear chromatin condensation, and evident structural disorganization. Treatment with WSMoL (Figure 6d) resulted in a reduced number of epithelial nuclei, further indicating structural disruption.

## 3. Discussion

The use of biorationals for insect control has grown substantially in recent decades, driven by heightened concerns about environmental sustainability [22]. Studies on alternative forms of biological control for managing *A. diaperinus* evaluated the use of entomopathogenic fungi [23] and essential oils [24,25]. However, compared to other insect pests, the search for natural compounds to control the lesser mealworm is still underdeveloped and has much room for further exploration. Furthermore, to the best of our knowledge, there is a gap in the scientific literature regarding the toxicity of lectins to *A. diaperinus*, with only one previous study reporting the effects of a lectin (MvRL) isolated from the rhizomes of *Microgramma vacciniifolia* (Langsd. & Fisch.) Copel. (Polypodiaceae), as well as crude preparations (extract and fraction) containing it, on the survival and enzymatic activities of the adults and larvae of this insect [26].

The procedure previously established for isolating WSMoL [14] involves protein extraction in distilled water (to obtain AE), protein precipitation using ammonium sulfate (yielding PF), and chromatography on a chitin column, recovering the purified lectin. The success of reproducing this procedure was confirmed by observing an increase in the specific hemagglutinating activity of the preparations at each step. Since WSMoL was found to be 842 and 410 times purer than AE and PF, respectively, the lectin can be considered effectively purified. Moura et al. [27] demonstrated that WSMoL is a 60 kDa chitin-binding acidic protein with an isoelectric point of 5.5.

Our study began by addressing the following question: would the contact with crude or purified preparations containing WSMoL reduce the survival of larvae and adults of *A. diaperinus*? The method used here is commonly employed to assess the acute effects of candidate insecticides as the insects come into contact with the solution, and the results are observed 48 h later [26]. Adult insects aged 20–30 days were used in our trials, as this stage corresponds to the period when they achieve full reproductive competence [2,4,5]. AE and PF exhibited contact toxicity effects on the adults, with PF proving to be more toxic than AE, while WSMoL showed no insecticidal effect (Figure 1a–c). A previous study indicated that, in addition to proteins, AE contains ellagic acid, flavonoids (including rutin), tannins, saponins, phenylpropanoids, alkaloids, and reducing sugars, whereas PF contains only reducing sugars [28]. Considering these findings and the inactivity of WSMoL, it can be inferred that *M. oleifera* seeds contain other proteins capable of exerting contact toxicity on *A. diaperinus* adults, as well as secondary metabolites that may be involved in this effect. An ethanolic extract from the leaves of *Ageratum conyzoides* L. (Asteraceae) containing alkaloids, steroids, terpenoids, tannins, and flavonoids was toxic to *A. diaperinus* adults [29]. Steroids, triterpenes, alkaloids, tannins, and flavonoids were also present in extracts (prepared in ethyl acetate, acetic acid, or hexane) from *Myrcia oblongata* DC. (Myrtaceae) leaf that caused the mortality of *A. diaperinus* adults [30]. Our results stimulate future evaluations on the contact toxicity on *A. diaperinus* of other compounds detected in AE, such as rutin and ellagic acid, which have both already been reported as insecticidal agents [31,32].

Regarding the fourth instar larvae, AE, PF, and even WSMoL exhibited contact toxicity (Figure 1d), suggesting that WSMoL could be an active principle responsible for this effect. Differences in the body structure between life stages may account for the varying effects of WSMoL on adults and larvae. Indeed, the tegument of the larvae is less sclerotized than that of adults [23], which may facilitate the penetration of this lectin. In contrast, for adults, the secondary metabolites in the extract may have penetrated more easily, and the greater diversity and quantity of proteins in PF could have compensated for the more challenging penetration of WSMoL. Similarly, the lectin MvRL showed contact toxicity only to *A. diaperinus* larvae and not to adults [26]. It is noteworthy that both WSMoL and MvRL are chitin-binding lectins [14,33], and it has been demonstrated that WSMoL can disturb the sclerotized shell of *Ae. aegypti* eggs, leading to the extensive degeneration and penetration of this protein inside the embryos [17].

To assess whether the survival and feeding preferences of adult insects can be altered by the presence of formulations containing WSMoL in the feeding substrate, the lectin preparations were incorporated into an artificial diet in which the adult insects were maintained for an extended period (21 days), and ingestion toxicity was measured. The data presented above (Figure 2) showed that, with respect to mortality induction, AE and PF were effective, whereas WSMoL did not reduce the insects’ survival rates. In this test, AE exhibited greater activity than PF, which aligns with the fact that pure WSMoL is not active. This is further supported by the high purity level of WSMoL compared to AE and PF. A study on the ingestion toxicity effects of AE and WSMoL on *Sitophilus zeamais* Motsch (Coleoptera: Dryophthoridae) adults similarly reported that the extract exhibited toxic effects, but that WSMoL was less active [28]. A tannin-rich extract from *Urtica dioica* L. (Urticaceae) leaves caused the mortality of *Spodoptera littoralis* (Boisduval) (Lepidoptera: Noctuidae) adults when tested in an artificial diet [34].

Although the results regarding the survival rates differed, all of the samples negatively affected the nutritional parameters of the adults (Figure 3a–c and Figure 4) and had a deterrent effect (Figure 3d–f), which probably caused the insects to lose weight (impaired conversion efficiency). However, this nutritional damage did not appear to be sufficient to cause the mortality of the insects within the 21-day evaluation period. Therefore, it is likely that components present solely in the extract were decisive for a more rapid reduction in the survival rate. On the other hand, the ingestion of WSMoL may affect the insects more gradually and, over a longer period, impact both survival and even reproductive fitness. In general, the effects of AE, PF, and WSMoL on nutritional parameters did not exhibit a dose–response relationship. This may be due to the saturation of bioactive components at the target sites, the activation of compensatory mechanisms by the insects, or the degradation/inactivation of active compounds at higher doses. Additionally, variations in the insects’ metabolic capacity, such as differences in detoxification pathways or enzymatic activity, may influence the observed responses. The ingestion toxicity assay was not conducted with *A. diaperinus* larvae due to their known cannibalistic behavior [26], which makes it impractical to maintain the experiment for an extended period, such as 21 days.

Bartling et al. [35] emphasized that insecticidal effects can also result from the influence of sublethal doses on insect physiology, leading to disrupted development and reproduction, caused by a mixture of factors including feeding, mobility, navigation, oviposition, and other behavioral elements. Müller et al. [36] found that sublethal exposure to pyrethroids in *Phaedon cochleariae* (Fabricius) (Coleoptera: Chrysomelidae) affected not only the immediate generation, but also the two following generations, resulting in changes to offspring development, chemical phenotype, and antenna symmetry. Therefore, our findings provide new opportunities for research on the effects of WSMoL antinutritional properties on the reproductive and behavioral fitness of *A. diaperinus*. Secondary metabolites from plants have also been reported as deterrent agents. For example, β-damascone, a ketone found in essential oils, showed a feeding deterrent effect for *A. diaperinus* adults when incorporated into a diet composed of oat flakes [37]. Also, flavonoids have been reported as feeding deterrent agents on insects [38].

In order to verify if, in addition to the deterrent action, the samples had caused any damage to digestion and nutrient absorption, the activity of trypsin-like enzymes in the guts of the adults who had ingested AE, PF, or WSMoL was assessed. Most insects use trypsin to digest dietary proteins, and it is widely present in the digestive tracts of insects from various orders and with diverse feeding habits [39]. The results highlight the strong inhibitory effect of WSMoL (Figure 5), a component likely responsible for the effects also observed for the AE and PF treatments. This inhibition of protein digestion, with a consequent reduction in the availability of essential amino acids, may have impacted the reduction in biomass, and may also be reflected in the difficulty of converting the diet into weight. Despite this, the absence of mortality in the group treated with WSMoL could be due to a compensation of the inhibition of trypsin activity, at least in part, by the secretion of other types of proteases [12]. The in vitro modulatory effect of WSMoL on insect digestive enzymes has been reported. However, this lectin was shown to stimulate the trypsin-like activity from gut extracts of *Ae. aegypti* [15] and *S. zeamais* [28].

Midgut digestive cells play a role in enzyme secretion, nutrient absorption and storage, and in fluid and ion transport [40]. The midgut epithelium of beetles undergoes continuous remodeling through the degeneration and/or extrusion of individual cells or cell fragments. These losses are subsequently replenished by regenerative cell pouches that restore the epithelium in actively feeding individuals [41]. Lectins have been shown to induce various structural changes in the insect midgut, including the disruption of the peritrophic matrix, brush border, and secretory cell layer; the induction of apoptosis and oxidative stress; and interference with nutrient absorption and transport proteins. These alterations impair the digestive and regenerative functions of the midgut [12]. Although a strong disorganization of the midgut epithelium was not observed, WSMoL and the other preparations containing it produced some degree of cell deformation, vacuolization, and a reduction in activity (Figure 6), which may also be linked to the antinutritional effect. These data address the final research question of our study, demonstrating that the ingestion of AE, PF, and WSMoL by adult insects can modulate trypsin-like enzyme activity and cause structural damage in the midgut.

## 4. Conclusions

Together, the data obtained in the present study show that all lectin preparations from *M. oleifera* seeds can be explored in the future for the development of sustainable products for the control of the lesser mealworm. For example, liquid formulations can be developed by exploring the contact toxicity of AE, PF, and WSMoL, while granules can be produced for the control of adults, based on the lethal and/or antinutritional effects of these preparations. It would also be interesting to assess the sub-lethal effects on *A. diaperinus* larvae and adults which survived to observe their lifespan, fertility, and fecundity, for example. It is important to emphasize that replacing synthetic compounds with biorational insecticides can only be achieved if the potential unintended effects of the natural agents are also considered and studied [42]. In this sense, WSMoL has been studied regarding its toxicity to non-target organisms, with lethal and sublethal doses already determined for fish (*Danio rerio* larvae) [43] and mammals (*Mus musculus*) [44]. The toxicity assessment panel has been expanded by our research group and is essential for strategies to be employed in the development of products and the definition of field application methods.

## 5. Materials and Methods

### 5.1. Insects

The insects were hand-collected on 15 March 2023 from poultry beds in Paudalho, Pernambuco, Brazil, and transported to the insectary of the *Laboratório de Bioquímica de Proteínas* (BIOPROT) at the *Departamento de Bioquímica*, *Universidade Federal de Pernambuco* (UFPE). Upon arrival, the insects were placed in plastic boxes (38 cm × 17 cm × 10 cm) containing a diet consisting of a 1:1 (*w*/*w*) mixture of oat flakes (Quaker Oats Company, Chicago, IL, USA), crushed rabbit food (Presence, Paulínia, São Paulo, Brazil), and small pieces of broiler litter. Larvae, pupae, and adults were separated into different non-transparent plastic boxes containing a nylon screen and were monitored every day. As new larvae, pupae, and adults emerged, they were transferred to other separate boxes with the same diet. The maximum number of insects per cage was 500. The colonies were maintained in the dark at 25 ± 2 °C and 75–80% relative humidity.

### 5.2. Moringa oleifera Seed Preparations

The seeds of *M. oleifera* were gathered at the UFPE campus in Recife, Pernambuco, Brazil. A reference specimen (no. 73345) is stored at the Dárdano de Andrade Lima Herbarium of the *Instituto Agronômico de Pernambuco*. The current study was registered (no. A68A2BA) with the *Sistema Nacional de Gestão do Patrimônio Genético e do Conhecimento Tradicional Associado* (SisGen), and the plant collection was authorized (no. 72024) by the *Instituto Chico Mendes de Conservação da Biodiversidade* (ICMBio) of the Brazilian Ministry of Environment. The preparations containing the lectin WSMoL were obtained as outlined by Coelho et al. [14]. The seeds were ground into powder using an LQL-4 industrial blender (Metvisa, Brusque, Brazil) for 5 min, and the resulting flour was suspended in distilled water for 16 h at 4 °C with continuous stirring. After filtration through cotton gauze and centrifugation (3000× *g*, 15 min, 4 °C), the aqueous extract (AE) was obtained. To obtain the protein-rich fraction (PF), the extract was treated with ammonium sulfate at 60% saturation [45] for 4 h. After this, centrifugation (3000× *g*, 15 min, 4 °C) was performed, and the precipitate was collected and dialyzed against distilled water (8 h, with three liquid changes), resulting in the PF [14]. For WSMoL isolation, the PF was applied to a chitin (Sigma-Aldrich, St. Louis, MO, USA) column equilibrated with 0.15 M NaCl. WSMoL was eluted from the column using 1.0 M acetic acid, dialyzed against distilled water (8 h, with three liquid changes), and then lyophilized [14].

The protein concentration was assessed following the method of Lowry et al. [46], utilizing a standard curve of bovine serum albumin (31.25–500 µg/mL). Lectin activity was evaluated through the hemagglutination assay (HA) as outlined by Procópio et al. [47], employing glutaraldehyde-fixed rabbit erythrocytes [48]. The erythrocyte collection procedure was approved by the Ethics Committee on Animal Use at UFPE (process no. 23076.033782/2015–70). Specific HA was defined as the ratio between HA and protein concentration (mg/mL).

### 5.3. Evaluation of Contact Toxicity on A. diaperinus Larvae and Adults

The evaluation of contact toxicity (short-term effects) was carried out following Zafeiriadis [49], with certain modifications as outlined by Santos et al. [26]. Initially, Petri dishes (90 mm × 15 mm) were each filled with 250 µL of AE (2.0, 4.0, 6.0, 8.0, or 10.0 mg/mL of protein), PF (2.0, 4.0, 6.0, 8.0, or 10.0 mg/mL of protein), WSMoL (0.2, 0.4, 0.6, 0.8, and 1.0 mg/mL), or distilled water (control). The samples were evenly spread across the surface of the plate for 30 s. Subsequently, 10 unsexed larvae (fourth instar) or 10 unsexed adults (20–30 days old) were placed onto each plate, which was subsequently covered with cling film. Mortality was recorded after 48 h of incubation in the dark at 25 ± 2 °C with 75–80% relative humidity. Insects were considered dead when they did not respond to a needle touch. Four independent trials were conducted (two for larvae and two for adults), with each trial performed in quintuplicate.

### 5.4. Ingestion Toxicity Assay in Adults

The toxicity by ingestion of unsexed adult insects (20–30 days old) was assessed following the methodology of Rice and Lambkin [50]. Initially, the insects were divided into groups of 10 individuals and weighed. In Petri dishes (90 mm × 15 mm), 1 mL of AE (2.0, 6.0, and 10.0 mg/mL of protein), PF (2.0, 6.0, and 10.0 mg/mL of protein), WSMoL (0.2, 0.4, 0.6, 0.8, and 1.0 mg/mL), or distilled water (control) was mixed with 1 g of oat flakes and 0.1 g of rabbit feed. The mixture was then dried in an oven at 56 °C for 120 min. Subsequently, a group of 10 adults was placed on each dish, and the experiments were incubated in the dark at 25 ± 2 °C and 75–80% humidity. Two separate trials were conducted, each in quintuplicate, totaling ten replicates for each concentration or control. Mortality rates were recorded after 21 days.

The feeding deterrence index (FDI) was computed using the formula: FDI (%) = 100 × (A − B)/A, where A represents the amount of food consumed by insects in the control assay and B is the amount ingested in the test [51]. According to the FDI, the sample was categorized as promoting no feeding deterrence (FDI < 20%), a weak feeding deterrence (20% ≤ FDI < 50%), a moderate feeding deterrence (50% ≤ FDI < 70%), or a strong feeding deterrence (FDI ≥ 70%) [21].

The following nutritional indices were calculated: relative consumption rate = C/(D × days), where C is the ingested mass (mg) and D is the initial biomass (mg) of the insects; relative biomass gain rate = E/(D × days), where E is the biomass (mg) gained or lost by the insects; and the efficiency of the conversion of ingested food = E/(C × 100).

### 5.5. Effects on the Activity of Trypsin-like Enzymes

To gather gut extracts, adult *A. diaperinus* that survived the toxicity ingestion assays with AE (2.0, 6.0, and 10.0 mg of protein/g), PF (2.0, 6.0, and 10.0 mg of protein/g) or WSMoL (0.2, 0.6, and 1.0 mg/g) were immobilized by cooling at 4 °C for 10 min. The guts were then carefully dissected using a needle and homogenized in 1 mL of 0.1 M Tris-HCl (pH 8.0) containing 0.02 M CaCl_2_ and 0.15 M NaCl, using a 2 mL glass tissue grinder. The homogenates were then centrifuged at 9000× *g* for 15 min at 4 °C, and the resulting supernatants (gut extracts) were collected for protein content analysis according to Lowry et al. [46].

The determination of trypsin-like activity was carried out in 96-well microtiter plates following the method of Kakade et al. [52]. The gut extracts (10 µL) were incubated for 30 min at 37 °C with 5 µL of N-benzoyl-DL-arginine-β-nitroanilide (BApNA, 8 mM) in Tris buffer (185 µL). The trypsin-like activity was determined by measuring the absorbance at 405 nm. One unit of trypsin-like activity represented the amount of enzyme that hydrolyzed 1 µmol of BApNA per minute. Two independent experiments, each conducted in duplicate, were performed.

### 5.6. Midgut Histological Analysis

The toxicity assay through ingestion was conducted as outlined in Section 5.4 for the following treatments: AE (10 mg of protein/g), PF (10 mg of protein/g), and WSMoL (1 mg/g). After exposing the insects for 21 days, the guts from the 10 survived insects per treatment were dissected using a stereomicroscope and microdissection scissors. The dissected guts were then fixed in a 4% paraformaldehyde solution. Following fixation, the samples were removed from the solution, rinsed three times with distilled water, and dehydrated through a graded ethanol series (ranging from 30% to 99%). The tissues were subsequently embedded in Historesin (Leica, Solms, Germany) for 24 h. Next, the samples were sectioned into 5 µm slices using a microtome (Reichert Jung 2050). The resulting sections were stained with Hematoxylin and Eosin (HE) to assess the morphological changes and cellular components. After staining, the sections were examined under an optical microscope and photographed using an Olympus BX53 microscope equipped with an Olympus DP 73 digital camera (Olympus Optical Co., Tokyo, Japan).

### 5.7. Statistical Analysis

All data are presented as the mean ± standard error of the mean (SEM). The Shapiro–Wilk normality test confirmed that the data followed a normal distribution. Statistical analysis was performed using Prism 8.01 software (GraphPad, La Jolla, CA, USA), and employing one-way analysis of variance (ANOVA) followed by Tukey’s post-hoc test. A significance level of *p* < 0.05 was used for all statistical comparisons.

## Figures and Tables

**Figure 1 plants-14-00511-f001:**
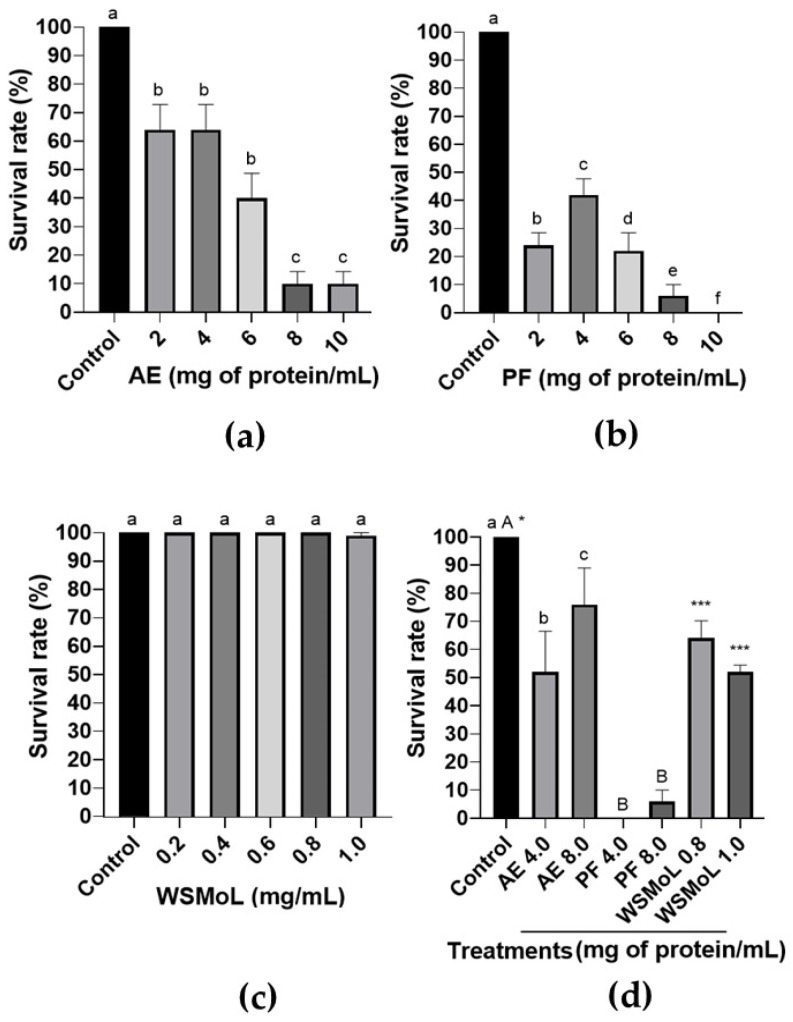
Contact (short-term) toxicity exerted by aqueous extract (AE), protein fraction (PF), and water-soluble lectin (WSMoL) from *M. oleifera* seeds on *A. diaperinus*. The survival rates of adults that were in contact with AE (**a**), PF (**b**), and WSMoL (**c**) are represented. Different letters indicate significant differences (*p* < 0.05) between the treatments. (**d**) Survival rates of larvae that were in contact with AE, PF, or WSMoL. Data were analyzed by one-way ANOVA followed by Tukey’s test. Different lowercase letters indicate significant differences (*p* < 0.05) between AE treatments and control. Different uppercase letters indicate significant differences (*p* < 0.05) between PF treatments and control. Different numbers of asterisks indicate significant differences (*p* < 0.05) between WSMoL treatments and control. In all graphs, the bars represent mean ± SEM.

**Figure 2 plants-14-00511-f002:**
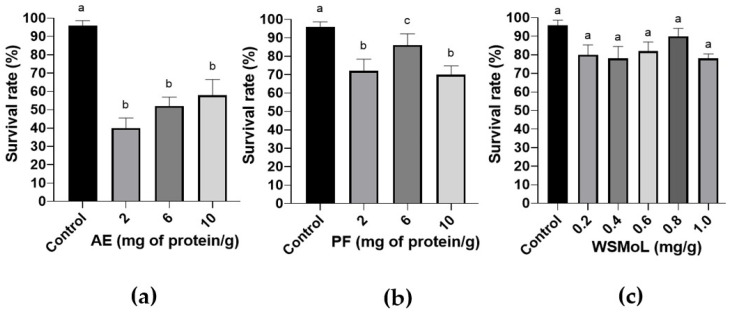
Evaluation of ingestion toxicity exerted by aqueous extract, AE (**a**); protein fraction, PF (**b**); or water-soluble lectin, WSMoL (**c**) from *M. oleifera* seeds on *A. diaperinus* adults. Different letters indicate significant differences (*p* < 0.05) between the treatments according to one-way ANOVA followed by Tukey’s test. The bars represent mean ± SEM.

**Figure 3 plants-14-00511-f003:**
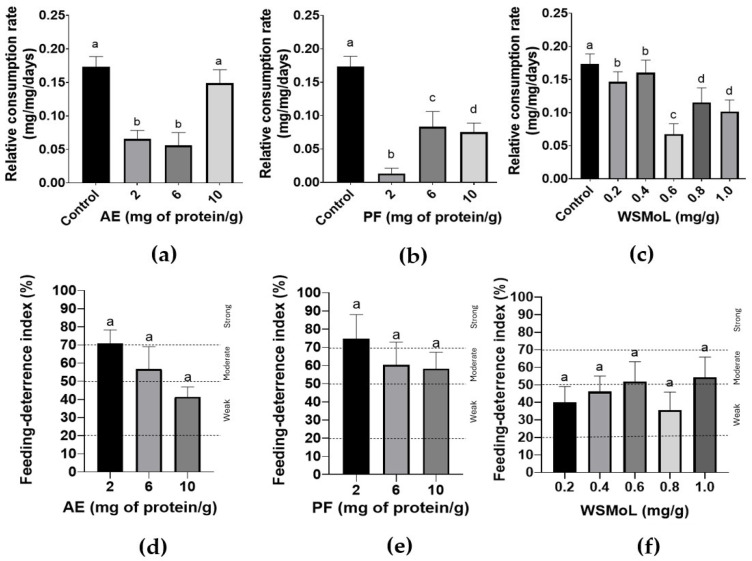
Relative consumption rate (**a**–**c**) and feeding deterrence indices (**d**–**f**) determined in the ingestion toxicity assay on *A. diaperinus* adults for treatments with aqueous extract, AE (**a**,**d**); protein fraction, PF (**b**,**e**); and water-soluble lectin, WSMoL (**c**,**f**) from *M. oleifera* seeds. Different letters indicate significant differences (*p* < 0.05) between the treatments, according to one-way ANOVA followed by Tukey’s test. The bars represent mean ± SEM. The dashed lines indicate the classification intervals of the deterrent effect according to the index values [21].

**Figure 4 plants-14-00511-f004:**
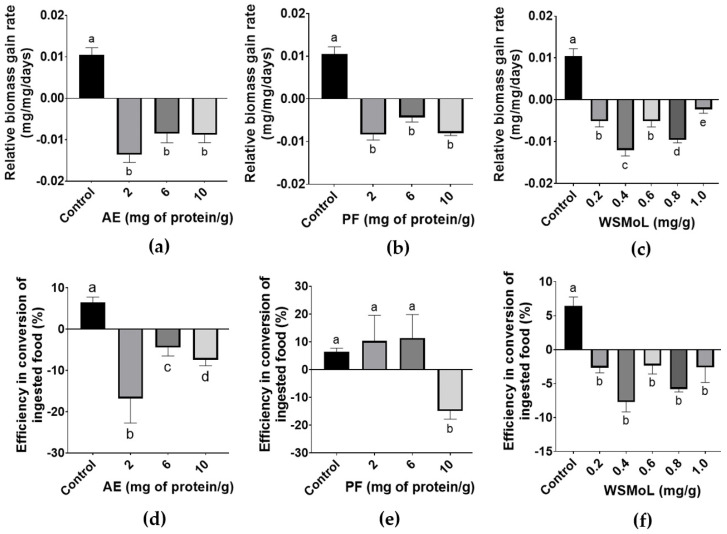
Relative biomass gain rate (**a**–**c**) and efficiency in the conversion of ingested food (**d**–**f**) determined in the ingestion toxicity assay on *A. diaperinus* adults for treatments with aqueous extract, AE (**a**,**d**); protein fraction, PF (**b**,**e**); and water-soluble lectin, WSMoL (**c**,**f**) from *M. oleifera* seeds. Different letters indicate significant differences (*p* < 0.05) between the treatments, according to one-way ANOVA followed by Tukey’s test. The bars represent mean ± SEM.

**Figure 5 plants-14-00511-f005:**
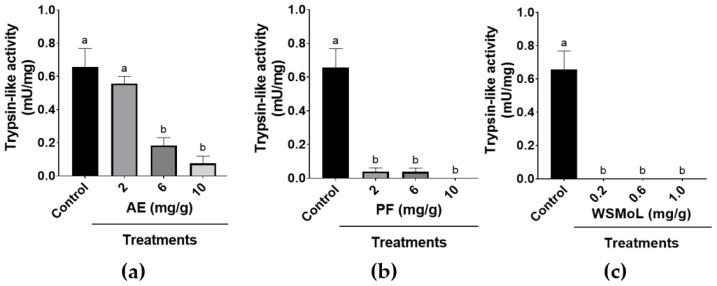
Trypsin-like activity in gut extracts from *A. diaperinus* adults that survived after exposure for 21 days to a diet containing aqueous extract, AE (**a**); protein fraction, PF (**b**); or water-soluble lectin, WSMoL (**c**) from *M. oleifera* seeds. Different letters indicate significant differences (*p* < 0.05) between the treatments, according to one-way ANOVA followed by Tukey’s test. The bars represent mean ± SEM.

**Figure 6 plants-14-00511-f006:**
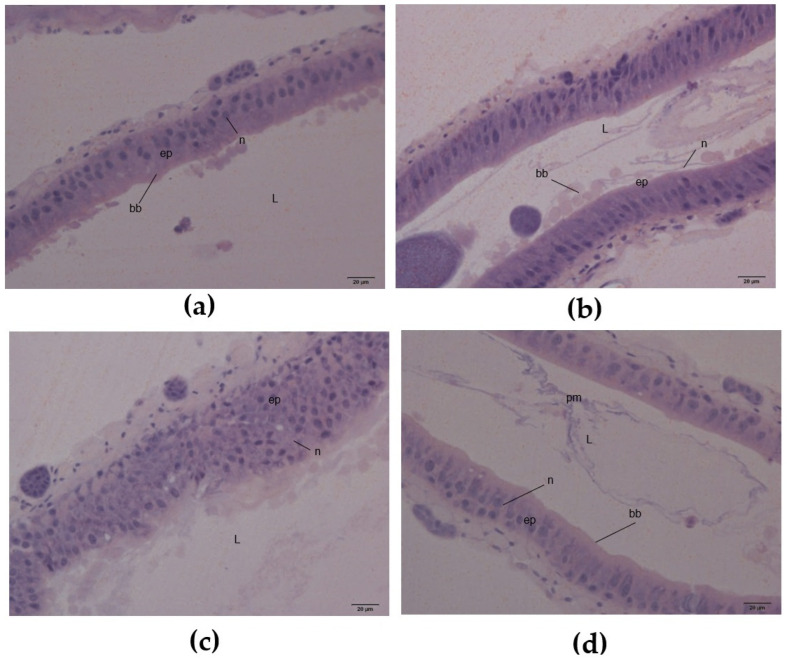
Midgut histological sections of *A. diaperinus* adults fed with a control diet (**a**), or a diet containing aqueous extract (AE) at 10 mg/g (**b**), protein fraction (PF) at 10 mg/g (**c**), or water-soluble lectin (WSMoL) at 1 mg/g (**d**) from *M. oleifera* seeds for 21 days. Lumen (L), epithelium (ep), digestive cell nucleus (n), brush border (bb), peritrophic matrix (pm), and basal cell nucleus. Stained with hematoxylin and eosin. Scale bar: 20 µm.

## Data Availability

The raw data supporting the conclusions of this article will be made available by the authors on request.

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
