# Peer review of "Insecticidal Activity of Lectin Preparations from Moringa oleifera Lam. (Moringaceae) Seeds Against Alphitobius diaperinus (Panzer) (Coleoptera: Tenebrionidae)"

_plants, 2025, doi:10.3390/plants14040511_

Round 1
Reviewer 1 Report
Comments and Suggestions for Authors
Abstract:
In abstract, WSMoL exhibited anti-nutritional effects, including a 94% reduction in trypsin-like activity, but did not cause lethality. The impaired trypsin activity may not directly cause lethal effects but may lead to insect mortality.
Introduction
It will be preferable to add a paragraph on lectins, highlighting the distinction between lectins in general and chitin-binding lectins. Chitin binding lectin which specifically recognize and bind to chitin polymers and may have the ability to disrupt the chitin metabolism in insects.
Results:
Why, WSMOL showed decrease in trypsin activity but in ingestion toxicity it showed weak or moderate effects?
Why, WSMoL affecting larvae but not adult insects?
Discussion
The discussion section is lengthy and can be made more concise. A more concise version would provide a focused analysis, highlighting key points and avoiding unnecessary details.
Overall conclusion comments:
The manuscript is well-written overall; however, it is recommended to address the comments provided.

Reviewer 2 Report
Comments and Suggestions for Authors
I reviewed the manuscript plants-3431987: I found the approach and the objective very interesting.
However there are few items that must be improved. In particular:
1. It would be important to provide additional information on the life cycle (and its impact) of the target insect pest in the introduction;
2. The Mat. and Methods are very often not very clear: my suggestion is to find a mother tongue English person to support the English version and to improve the terminology and the description of the experiments (see my comments in the PDF);
3) there also some question marks (see my comments in the PDF) that should receive an answer or must be reconsidered for further investigations (working with adults, you did not consider te importance of the gender: males and females generally are feeding in a different ways);
4) Improve the discussion: when you are describing the achieved results, it would make sense to add the referred number of the graphic or the table, to help the reeder to better understand the data you are showing ant the conclusions you are achieving.
5) According to some of the graphs, the response of the different formulations tested was not following a trend in line with their respective concentrations. I did not find a clear explanation of these data both in the Results and in the Discussion chapters.

Definitely, the English should be improved: it would be important to receive a support of a mother tongue English person to improve the quality of the text: since the Authors have done an important work (evaluating the effects of the extracts in different ways): it is important to provide correct tools to the reeders to better understand the manuscript.
Reviewer 3 Report
Comments and Suggestions for Authors
It has been a real pleasure to read the research article titled “Insecticidal Activity of Lectin Preparations from Moringa oleifera Lam. (Moringaceae) Seeds against Alphitobius diaperinus Panzer (Coleoptera: Tenebrionidae)”. The work is well-written, clearly conceived and executed, useful, and innovative. Indeed, there are limited studies on A. diaperinus as a poultry pest, as it is generally more considered a food/feed source, so this research was needed.
The work includes all the M. oleifera extracts preparation, A. diaperinus mortality by contact and ingestion, antinutritional effects, enzymatic tests, and histology analysis. Finally, it clearly underlines the need for further toxicity trials on non-target organisms.
My few minor remarks and suggestions are reported in the attached PDF.
Congratulations!
